# Growth and Physiological Characteristics of Strawberry Plants Cultivated under Greenhouse-Integrated Semi-Transparent Photovoltaics

**DOI:** 10.3390/plants13060768

**Published:** 2024-03-08

**Authors:** Theodoros Petrakis, Paraskevi Ioannou, Foteini Kitsiou, Angeliki Kavga, George Grammatikopoulos, Nikos Karamanos

**Affiliations:** 1Department of Agriculture, University of Patras, 26504 Patras, Greece; tpetrakis@ac.upatras.gr; 2Biochemistry, Biochemical Analysis & Matrix Pathobiology Research Group, Laboratory of Biochemistry, Department of Chemistry, University of Patras, 26504 Patras, Greece; ioannouevi@icloud.com (P.I.); n.k.karamanos@upatras.gr (N.K.); 3Laboratory of Plant Physiology, Department of Biology, University of Patras, 26504 Patras, Greece; kitsioufot@gmail.com (F.K.); grammati@upatras.gr (G.G.); 4Foundation for Research and Technology-Hellas (FORTH)/Institute of Chemical Engineering Sciences (ICE-HT), 26504 Patras, Greece

**Keywords:** *Fragaria x ananassa* Duch., shading, agrivoltaics, crop performance, total phenolic content, antioxidant activity

## Abstract

The integration of semi-transparent photovoltaics into the roof of greenhouses is an emerging technique used in recent years, due to the simultaneous energy and food production from the same piece of land. Although shading in many cases is a solution to maintain the desired microclimate, in the case of photovoltaic installations, the permanent shading of the crop is a challenge, due to the importance of light to the growth, morphogenesis, and other critical physiological processes. In this study, the effect of shade from semi-transparent photovoltaics on a strawberry crop (*Fragaria x ananassa* Duch.) was examined, in terms of growth and quality (phenolic and flavonoid concentration of fruits). According to the results, in non-shaded plants, there was a trend of larger plants, but without a significant change in leaf number, while the total number of flowers was slightly higher at the end of the cultivation period. Moreover, it was found that the percentage change between the number of ripe fruits was smaller than that of the corresponding change in fruit weight, implying the increased size of the fruits in non-shaded plants. Finally, regarding the antioxidant capacity, it was clearly demonstrated that shading increased the total phenolic content, as well as the free-radical-scavenging activity of the harvested fruits. Although the shading from the semi-transparent photovoltaics did not assist the production of large fruits, it did not affect their number and increased some of their quality characteristics. In addition, the advantageous impact of the semi-transparent photovoltaics in the energy part must not be neglected.

## 1. Introduction

Strawberry plants generally thrive in well-drained soil and temperate climates. The Mediterranean climate is well-suited for strawberry cultivation due to its mild, wet winters and hot, dry summers. Strawberry plants are often grown in greenhouses either for protection from adverse environmental conditions or, more commonly, to extend the growing season [1]. Light and temperature inside the greenhouse are the most influencing factors that affect the growth of strawberry plants [2]. The optimum light conditions vary according to the time of year and the growth stage of the plant. Light intensity is important for the growth, morphogenesis, and other physiological processes of plants by affecting the rate of photosynthesis, the number of leaves, the stem length, the branching, the commencement of flowering, and the fruit development and ripening [3,4,5]. In particular, low light intensity can reduce the rate of photosynthesis in plants, resulting in limited growth and lower yields [6]. The effects of reduced light intensity on the quality of harvested fruits are also important because strawberries have a variety of secondary metabolites, like phenolics and flavonoids. These metabolites serve as plant tissue defense against biotic and abiotic stresses, preventing cellular damage caused by reactive oxygen species [7].

Fruit berries, especially strawberries, are well-known sources of bioactive compounds (BACs). The major components of BACs in strawberries are phenolic compounds, which include phenolic acids (such as hydroxybenzoic and hydroxycinnamic acids), flavonoids (anthocyanins), flavonols (quercetin, kaempferol, myricetin), and flavanols (catechins and epicatechin) [8,9]. These components along with ascorbic acid are the most abundant antioxidants present in strawberries. Antioxidants work by donating hydrogen to free radicals, scavenging them, and generating more stable antioxidant radicals [10].

Shading in greenhouses protects against excess solar radiation during hot periods while also posing challenges during periods of limited sunshine and low temperatures [11]. The fluctuations of environmental conditions like these are relevant in Mediterranean regions. In the summer and in regions such as Greece, the temperature rises dramatically inside a greenhouse, with its values approaching 60 °C for extended periods (often from May to October), causing severe crop issues. Shading prevents the overheating of plants and reduces evapotranspiration by reducing incident light intensity and, consequently, lowering the temperature inside the greenhouse. Yet, shading is generally considered a limiting factor for growth and yields [12,13].

Aside from conventional greenhouse shading solutions such as plastic nets [14,15,16,17] and screens [18,19], one approach that has seen significant expansion and application in recent years is the installation of photovoltaic modules on the greenhouse’s roof. Solar radiation is the most important parameter in satisfying production performance because photosynthesis is a biological process that significantly depends on sunlight. Despite that, the strength of an energy production system using solar radiation, such as a photovoltaic system, is primarily determined by the strength of the radiation incident on it. The requirements of both the crop and the solar system for sunlight make greenhouses the perfect structure, as they are placed in areas with a lack of surrounding impediments. Simultaneously, significant challenges have been encountered because of the increased demand for accessible land, both spatially and economically. Thus, using the same piece of land for both food and energy production appears to be a perfect solution, and this combination is also known as an agrivoltaic system [11,20,21].

According to the research findings, the employment of shade methods provides several advantages for preserving the greenhouse’s microclimate at desirable levels in warm and cold regions. Indicatively, in [22], it is stated that, on the one hand, shading can maintain the temperature of the greenhouse in a range of 5 to 10 °C lower than that of the outside air, while this reduction combined with the effect of a cooling system increases the relative humidity up to 20%. At the same time, solar radiation can be reduced by up to 50%. On the other hand, in cold climate regions, the application of shading materials can act as additional insulation, reducing heat leaks, with the temperature being maintained up to 5 °C higher than outside.

Regarding the use of semi-transparent photovoltaics in a greenhouse, in [23], it is stated that, by covering the roof of a polyethylene-covered greenhouse with 20% photovoltaics, the solar radiation can be reduced by 35–40% during clear days, while the temperature reduction with the use of photovoltaics was 1–3 °C. At the same time, it is reported that, with an annual energy production equal to 637 kWh, the payback period was 9 years.

Several studies on the impact of semi-transparent PV panels on plant development in greenhouse conditions have been conducted. According to the literature, shading impacts different plant species in different ways, with [24,25] focusing on tomato cultivation, [2,26] on strawberries, [27] on berries, and [28] on lettuce, while there are also numerous studies demonstrating different outcomes. More specifically, Ref. [29] investigated four shading levels, which were the result of four different rates of coverage, with percentages amounting to 0%, 15%, 30%, and 50%. The study was performed for tomato cultivation and for a greenhouse with its ridge pointed in the north–south direction. The results showed that the reduction of solar radiation, due to the increase in the degree of coverage, had negative effects, both on the quantity and the quality of the fruits.

This study aimed to investigate the effect of greenhouse-integrated semi-transparent photovoltaics’ shading on the parameters reflecting the size of the plant, the number of leaves, the flowers, the ripened fruits, and their weight. Moreover, the effect of shading on the antioxidant capacity (total phenolic content (TPC) and free radical scavenging activity) of the harvested fruits was examined.

## 2. Materials and Methods


### 2.1. Experimental Setup

#### 2.1.1. Greenhouse

The cultivation of the *Fragaria x ananassa* Duch. plants was conducted in the experimental greenhouse of the Plant Physiology Lab (Dept of Biology, University of Patras) located on the University campus (38°17′27.9″ N, 21°47′23.9″ E) in a distinct unit that measures 3.2 m in width and 16.5 m in length, from 15 December 2022 to 1 June 2023.

The greenhouse’s dimensions (gutter height of 3.22 m, ridge height of 3.9 m) are close to those of a real-scale greenhouse, with its type (even span) and east–west orientation, making it physically identical to a commercial greenhouse. The greenhouse is made up of four identical, distinct units. It is a high-tech greenhouse made of premium materials, and its frame is constructed of steel pieces with varying cross-sections. The cover material is glass, which is resistant to chemicals, high winds, and pollution. The greenhouse has a natural ventilation system with openings on the north and south inclined planes of the roof, as well as the side portions.

The construction unit used for the cultivation of strawberry plants is part of the overall greenhouse and, more specifically, is located on its northern side, with the area of covered ground equal to approximately 50 $m^{2}$. In this unit, natural ventilation is based only on windows on the northern sloped plane of the roof, due to the installation of photovoltaics on the southern sloped plane.

#### 2.1.2. Photovoltaic Modules

The photovoltaic modules are characterized as semi-transparent, with permeability to sunlight and especially the spectrum of photosynthetically active radiation (PAR) (400–700 nm) being particularly high, while at the same time, intense light scattering is observed on the surface that does not contain solar cells. This scattering has the advantage of avoiding strong shadows from any skeletal elements that may lie beneath them.

Each photovoltaic module has dimensions of 2.089 m long and 1.033 m wide, a thickness of 5.5 mm, and a weight of 28.6 kg. The solar cells lying between two sheets of glass and within an encapsulation layer of POE/EVA material are equal to 80. Their dimensions correspond to 166 mm long and 83 mm wide, making an active surface of approximately 1.1 $m^{2}$. Finally, their bifacial type increases energy production as it is reliant on light falling both on the top and bottom sides of each cell. The nominal maximum power of each photovoltaic module is equal to 250 Wp, while according to the dimensions of both the whole module and the solar cells, the shading formed corresponds to approximately 51%.

#### 2.1.3. Installation

Both in the arable unit and the unit next to it, twelve semi-transparent photovoltaics have been integrated. Their installation was performed by removing the old glass cover and replacing it by adding an aluminum bracket on the perimeter. The inclination angle of the photovoltaics is equal to ∼ 24°, so the inclination angle of the roof, while setting the units into the south sloped level of the roof increases the PVs’ energy production, particularly during the winter, when the Sun makes a smaller angle to the ground. Out of twelve solar panels, eight have been installed on the roof of the arable greenhouse unit (Figure 1a), while the rest have been installed on the roof of the adjacent unit (Figure 1b). Their long side follows the east–west ridge direction, with them covering approximately 67% and 34% of the south sloped roof, respectively.

### 2.2. Algorithm for Calculating the Shade

To determine the percentage of shading caused by the photovoltaics inside the greenhouse and on the ground, an algorithm based on the specific greenhouse was used, which enables the detection of shading caused by photovoltaics on the ground for each time of year and for a time step of 10 min. The algorithm is based on astronomical equations and theorems relating to the position of the Sun, and it presents the shade using vector analysis and geometry. The algorithm has been validated using radiation data collected from within the greenhouse, according to all of its geometric and geographical aspects [30]. The outputs of the algorithm are the shaded area of the greenhouse in $m^{2}$, the percentage of ground covered by shade (%), and two graphical representations of the greenhouse, PVs, and shade, one 3D and one 2D. Finally, by providing its coordinates, it is possible to obtain a result showing whether a surface point is inside or outside the shaded area.

In this study, just the percentage of the shaded region was used from the overall output of the algorithm, which was executed for two separate scenarios, one where the area of interest concerned the planting line A and one where the area of interest concerned the planting line B. It should be noted that, although the algorithm extracts accurate results based on the arrangement according to which the PVs are placed in the greenhouse, the PVs are considered opaque. To calculate the percentage of shading formed by the photovoltaics on each line, the theoretical shading caused by them must be included. This shading is a result of the relationship between the active surface (total surface of the solar cells) and the total surface of the photovoltaic module (Equation (Equation 1)).
(1)$$\mathrm{theoretical}\;\mathrm{shading}\;\mathrm{perc}.\;\left(\%\right)=\left(\frac{N\;\cdot \;E_{\mathrm{solar}\;\mathrm{cell}}}{E_{\mathrm{PV}\;\mathrm{module}}}\right)\times 100$$
where *N* is the number of solar cells in each PV module equal to 80, $E_{\mathrm{solar}\;\mathrm{cell}}$ is the surface of the solar cell equal to 0.01378 $m^{2}$, and $E_{\mathrm{PV}\;\mathrm{module}}$ is the total surface of each PV module equal to 2.158 $m^{2}$. According to the above, the resulting theoretical shading is equal to 51.08%, while this factor is also presented in the technical documentation accompanying the photovoltaic units, representing the semi-transparent nature of the modules. Thus, by multiplying the algorithm’s output by the theoretical percentage of shading created by the semi-transparent photovoltaics, we can compute the shading formed by the photovoltaics on the surface of each planting line.

The mean daily percentage of shaded surface based on the foregoing is provided in Figure 2, from the day the plants were planted until the last day of the experiment. In addition to the percentage of the shaded surface, the mean daily internal temperature is provided in Figure 2, with a loss of data for the period 13 January 2023 to 13 February 2023, due to temperature sensor failure.

### 2.3. Growth, Morphological, and Yield Measurements

Frigo plants were planted in two single rows with an approximately 1.5 m distance between rows. The length of each row was approximately 15 m; therefore, the spacing among the 20 plants used in each line was 75 cm. The soil was covered with black plastic mulch, and the plants were irrigated by a drip tube with a diameter of 20 mm under the mulch, delivering 2 L
$h^{-1}$ per dripper (Figure 3a,b). Each plant was equipped with one dripper, with the number of drippers corresponding to the number of plants. The irrigation frequency was scheduled every three days for the period from 15 December 2022 to 1 March 2023 and daily application until the end of the experiment, to ensure that the soil was at the field capacity. Measurements were taken every 15 days beginning on 15 February 2023, and the harvest on 15 March 2023 was extended by two days due to the unpredictable fruit development. The measurements performed on the crop concern (a) the plant size through three parameters: the plant size along the x-axis (along the planting line), the y-axis size (perpendicular to the planting line), and the z-axis (plant height); (b) the number of leaves; (c) the number of flowers; (d) the number of red (ripened) fruits; and (e) the weight of the ripe strawberries.

**Figure 3 plants-13-00768-f003:**
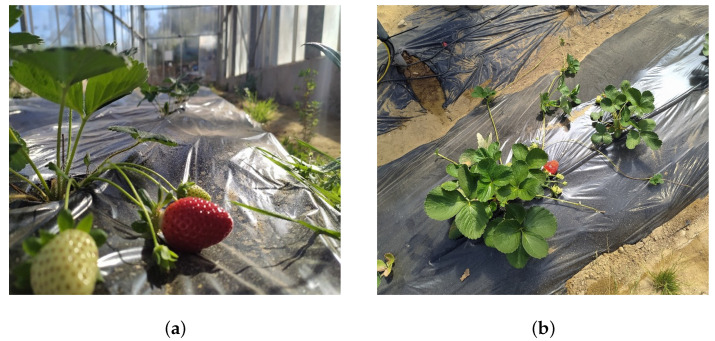
(**a**,**b**): Both subfigures represent strawberries’ cultivation in the greenhouse.

### 2.4. Sample Preparation and Extraction Methodology

The strawberry samples utilized in this study were stored at − 20 °C. For the extraction process, three strawberries from each line (denoted as A and B) were first allowed to thaw at room temperature (RT). Subsequently, each strawberry was weighed after careful removal of the stem. Next, each strawberry was homogenized to achieve uniformity in mass, and the resulting product was reweighed. Then, 1 g of the homogenized strawberry was transferred to a falcon tube, and 12 mL of 70:30 [methanol (MeOH)–2x distilled (2d) water] was added as the extraction solvent. The homogenate, along with the extraction solvent, was left for 1 h with frequent mild agitation in vortex to facilitate the extraction. Following this, centrifugation at 3000 rpm for 10 min at room temperature was performed to separate the supernatant, which was collected and used for chemical analyses.

### 2.5. TPC and DPPH Analyses

The TPC [31] in the strawberry extracts was obtained with Folin–Ciocalteu’s reagent (Panreac, Barcelona, Spain). In this study, each sample (0.125 mL) was combined with 0.75 mL of 2d water and 0.125 mL of the Folin–Ciocalteu reagent. The resulting solution was allowed to incubate for 6 min. Subsequently, 2 mL of a 5% (*w*/*v*) sodium carbonate solution was added, and the mixture was left to incubate for an additional 90 min at room temperature in the absence of light. The absorbance of the samples was measured at 760 nm using a TECAN photometer. The obtained results are expressed as mg of gallic acid equivalent (GAE) per g of strawberry, representing the average of three measurements for each sample in triplicate.

The free-radical-scavenging activity of the extract was evaluated using the 1,1-diphenyl-2-picryl-hydrazyl (DPPH) method [32]. More precisely, a 0.2 mM DPPH solution (Cayman Chemical, Ann Arbor, MI, USA) was prepared fresh in MeOH. In this study, each sample (0.02 mL) was combined with 0.180 mL of the freshly prepared 0.2 mM DPPH solution, and it was allowed to incubate for 30 min at room temperature in the absence of light. The absorbance of the samples was measured at 517 nm using a TECAN photometer. The obtained results are expressed as mg of L-ascorbic acid equivalent (AAE) per g of strawberry, representing the average of three measurements for each sample in triplicate.

### 2.6. Gallic Acid and L-Ascorbic Acid Calibration Curves

To obtain a 1% solution of gallic acid (GA) (10 mg
m$L^{-1}$), 0.15 g of GA (Sigma-Aldrich, St. Luis, MO, USA) was dissolved in 15 mL of 70:30 [methanol (MeOH)–2d water]. Subsequently, a standard gallic acid curve was established by diluting the standard solution of gallic acid to concentrations of 0.1, 0.5, and 1 mL in 70:30 [methanol (MeOH)–2d water]. The calibration curve was obtained using the TPC method, as described above. The equation obtained is represented by Equation (Equation 2) with a perfect linear regression of $R^{2}=1$.
(2)y=2.2657·x+0.1342

To obtain a 100 μg
m$L^{-1}$ solution of L-ascorbic acid (100 mg
m$L^{-1}$), 0.015 g of L-ascorbic acid (Acros Organics, Fair Lawn, NJ, USA) was dissolved in 150 mL of 2d water. Afterward, a standard L-ascorbic acid curve was established by diluting the standard solution of L-ascorbic acid to concentrations of 5, 10, 20, 30, 50, 75, and 100 μg
m$L^{-1}$ in 2d water. The calibration curve was obtained using the DPPH method, as described above. The equation acquired is represented by Equation (Equation 3) with a linear regression of $R^{2}=0.9424$.
(3)y=0.6858·x+4.21288

### 2.7. Statistical Analysis

The reported values for the growth, morphological, and yield measurements are expressed as the mean ± the standard error (SE).

The reported values are expressed as the mean ± the standard deviation (SD) of the experiments in triplicate. Statistically significant differences were evaluated using Tukey’s test to determine statistical differences between each data set of the two lines (A and B). Differences were considered statistically significant at the level of *p* ≤ 0.05, indicated by an asterisk (*). The statistical analysis and graphs were achieved using GraphPad Prism 8.0.1 (GraphPad Software, San Diego, CA, USA).

## 3. Results

### 3.1. Growth and Morphological Data

Using the three parameters relating to the size of each plant (*x*-, *y*-, and *z*-axis) and taking into account that the specified directions produce a Cartesian coordinate system in space, it is feasible to define a vector, $\vec{v_{i}}$, in space and for each plant, *i*, of the form (Equation (Equation 4)):(4)$$\vec{v_{i}}=\left(x_{i}\;\cdot \;\hat{x},y_{i}\;\cdot \;\hat{y},z_{i}\;\cdot \;\hat{z}\right)$$
where *x*, *y*, and *z* are the plant size along the *x*-, *y*-, and *z*-axis, respectively, and $\hat{x}$, $\hat{y}$, and $\hat{z}$ are the unit vectors in the directions along the planting line, perpendicular to the planting line, and the plant height, respectively. The magnitude of $\vec{v_{i}}$ will be a number (in cm) that, according to Equation (Equation 5), will define the size of the plant by integrating the above three parameters for the size of each plant.
(5)$$|\vec{v_{i}}|=\sqrt{x_{i}^{2}+y_{i}^{2}+z_{i}^{2}}$$

As shown in Figure 4, there was a trend of bigger plants in the line with the lower overall shading (line A), though the differences between the two lines were not statistically significant. The small, positive effect on the plant size for the plants in line A appeared 75 days from the beginning of the measurements (b.m.) when the cumulative shading difference between the two lines was obvious.

The mean number of leaves was almost identical between the two planting rows during the growth period (Figure 5a). The total number of leaves for each row at each measurement date was not affected by the fluctuations of radiation during this period (Figure 5c), resulting in a similar cumulative number of leaves until the end of the experiment (Figure 5e). The onset of flowering was simultaneous in the two rows, and the mean number of flowers per plant was similar between the planting rows during the experimental period (Figure 5b). As shown in Figure 5d, ascending phases of radiation positively affected the production of inflorescences in the corresponding planting row. Consequently, the final total number of inflorescences in line A was relatively higher than in line B, probably due to the coincidence of the longest ascending period of radiation with the peak of the flowering period in line A (Figure 5f).

In Figure 6, the number of ripened fruits and their weight are presented. There were no statistically significant differences in the number and weight of fruits per plant between the two planting rows for each measurement during the experimental period. However, at the peak of the fruiting period, a trend for heavier fruits per plant was obvious (Figure 6a,b). Shading reduced the number of fruits at the end of the experimental period by 20% and the weight of fruits by 45% (Figure 6c,d). The cumulative effect of shading was 10% and 35% for the number and weight of fruits, respectively (Figure 6e,f).

In Table 1, the number of leaves, flowers, ripened fruits, and weight of fruits at the final harvest are presented, as well as the percentage change between each parameter for lines A and B, together with the total shading posed by the photovoltaics on each planting line. The total shading values were calculated using the trapezoidal method with unit spacing, which computes the approximate integral of the data in Figure 2.

### 3.2. Total Phenolic Content and Free-Radical-Scavenging Activity

As mentioned in the Materials and Methods Section, samples from each line underwent the TPC method for the determination of their total phenolic content. The absorbance of each sample was measured at 760 nm, three samples in triplicate, and their TPC is given through the calibration curve of GA. The results are given in Figure 7. The difference between the two lines of strawberries was significant, as line B had a significantly higher TPC content than line A.

Moreover, three samples from each line underwent the DPPH method for the determination of their free-radical-scavenging activity/antioxidant activity. The absorbance of each sample was measured at 517 nm in triplicate, and their scavenging potential is given through the calibration curve of L-ascorbic acid. The results are presented in Figure 8. The difference between the two groups of strawberries was significant, as line B had 1.375 ± 0.2524 mg of AAE per g morethan line A.

## 4. Discussion

Shading from photovoltaic arrays on the roof of greenhouses can have a positive or negative effect on the growth of the cultivated plants, depending on the period during which the cultivation is carried out [11,33,34]. During the growth of strawberry plants in this study, due to diurnal and seasonal variations in light intensity and ambient temperature, plants may have experienced changes in incident radiation and, consequently, temperature within the greenhouse daily, which included either quite low temperatures during the winter period or quite high temperatures during the spring period. Therefore, shading from the PVs could theoretically have a negative effect on plant growth during the winter period, but mitigate the negative effect of high temperatures in the spring. At the same time, reduced light intensities are expected to have a negative effect on plant growth [22]. According to the results, there was a tendency of larger plants in the row with higher light intensities, while there was no effect of shading on the number of leaves throughout the experiment. This tendency of increased plant size was observed during spring, and therefore, this effect should be attributed to the slightly increased light intensity received by plants in row A. At the same time, the prevailing air temperature for the two rows was the same. Therefore, the effect of the photovoltaics on the shading was different between the two rows, but their effect on temperature was similar for all the plants. Consequently, any positive effect of the photovoltaics in mitigating high temperatures may contribute to the observed difference in plant size between the two rows, which otherwise could be smaller, if the more shaded plants had benefited from lower temperatures.

Shading did not affect the onset of flowering. The onset of flowering for each plant species is different and is usually determined by the prevailing photoperiod, i.e., the relative ratio of light–dark hours in a day and the progressive decrease or increase of this ratio [35]. This step in a plant’s life cycle is significant due to its adaptability to seasonal changes and its reproductive success [36]. The onset of flowering is not affected by light intensity [35], which was confirmed in the present study, with the onset of flowering being the same in both crop lines regardless of the shading amount. In previous studies, the shading did not affect the number of flowers for pepper plants [37] and strawberry plants [38]. However, in our results, the number of flowers for each crop line was lower compared to the other, after a preceding period of increasing shading. Since the total shading in line B was greater than the shading in line A, the total number of flowers for line A was slightly higher than for line B at the end of the cultivation period.

The yield of fruits was affected by shading in terms of fruit weight. The number of fruits was not affected, except in the last harvest, when the line with the higher shading produced 20% fewer fruits. At the end of the experiment, the total number of fruits was slightly lower under the higher shading treatment. However, the difference in fruit weight was significantly higher between the two lines, because the strawberries under higher light intensity were about 50% heavier during the last month of the experiment, that is the period with the peak of fruiting. As the number of fruits was similar between the lines, it is obvious that, under higher light intensity, bigger fruits were produced. Smaller fruits under shading could result in an altered secondary metabolites profile and, thus, change the fruits’ quality characteristics. In previous studies, the yield of tomatoes [39] was reduced only under the high level of shading (60%), while in a hydroponic culture of tomatoes, the reduction in the yield was gradually increased under increasing levels (15–50%) of shading [29]. On the other hand, the yield and quality of chili peppers improved under shading [37].

Shading is known to increase the TPC and antioxidant activity of many plants including coffee beans [40], perennial wall rocket [41], and sweet potato [42]. In our case, the results showed that the shading provided by the semi-transparent PV panels resulted in smaller fruits with a higher TPC and antioxidant activity.

Although the shading from the semi-transparent photovoltaics did not assist the production of large set fruits, even if the number was unaffected, the contribution of the shading resulted in higher quality fruits, while their wider impact in the energy part must not be neglected. Finally, it must be noted that the above results would probably be more intense if the comparison was between plants facing the increased shade from the photovoltaics and plants not receiving any additional shade at all.

In an era when efforts to create a complex between sustainable agriculture and RES are intensifying, the combination of greenhouse systems and semi-transparent photovoltaics can become a useful tool, not only for producing environmentally friendly energy, but also for qualitative optimization of greenhouse crops. The above example demonstrates how, through innovation, solutions may be discovered for two challenges: food security, on the one hand, and renewable energy production and climate change mitigation, on the other, for a more sustainable future.

## Figures and Tables

**Figure 1 plants-13-00768-f001:**
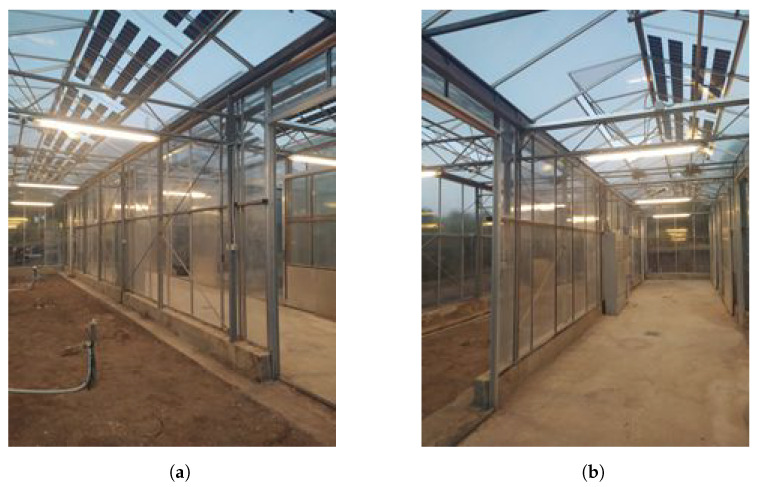
Photovoltaic modules integrated into the roof of the greenhouse: (**a**) Greenhouse arable construction unit. (**b**) Adjacent greenhouse unit.

**Figure 2 plants-13-00768-f002:**
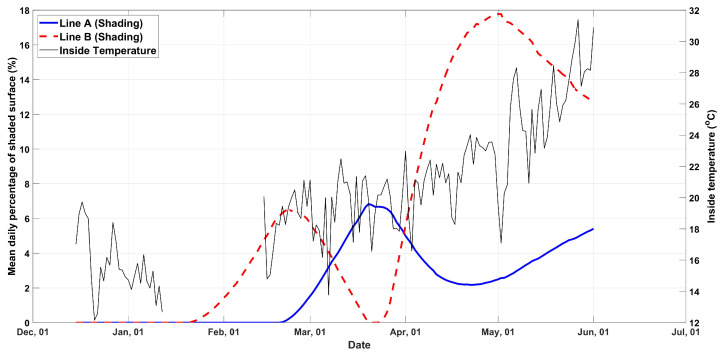
Mean daily percentage of shaded surface for planting lines A and B and internal temperature.

**Figure 4 plants-13-00768-f004:**
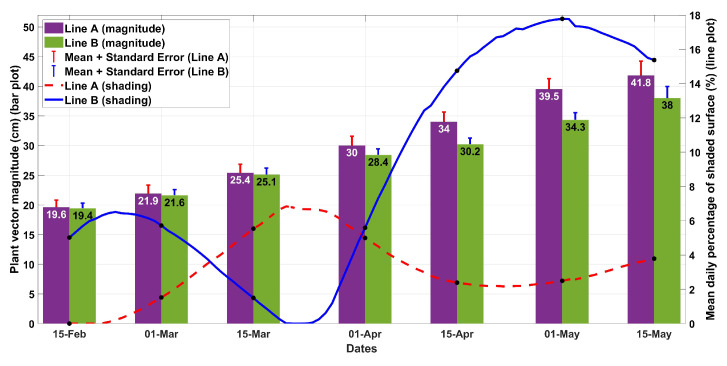
Mean plant vector magnitude for planting lines A and B (mean ± SE, n = 28) combined with the mean daily percentage of the shaded surface.

**Figure 5 plants-13-00768-f005:**
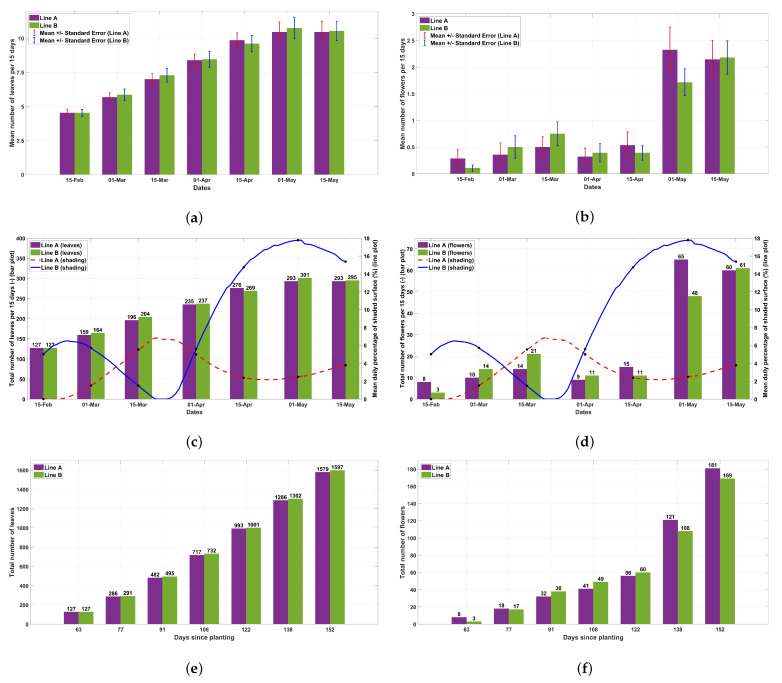
(**a**) Mean number of leaves during the experimental period (mean ± SE, n = 28). (**b**) Mean number of inflorescences during the experimental period (mean ± SE, n = 28). (**c**) Total number of leaves for each planting row combined with the mean daily percentage of the shaded surface. (**d**) Total number of inflorescences for each planting row combined with the mean daily percentage of the shaded surface. (**e**) Cumulative sum of the number of leaves for each planting row. (**f**) Cumulative sum of the number of inflorescences for each planting row.

**Figure 6 plants-13-00768-f006:**
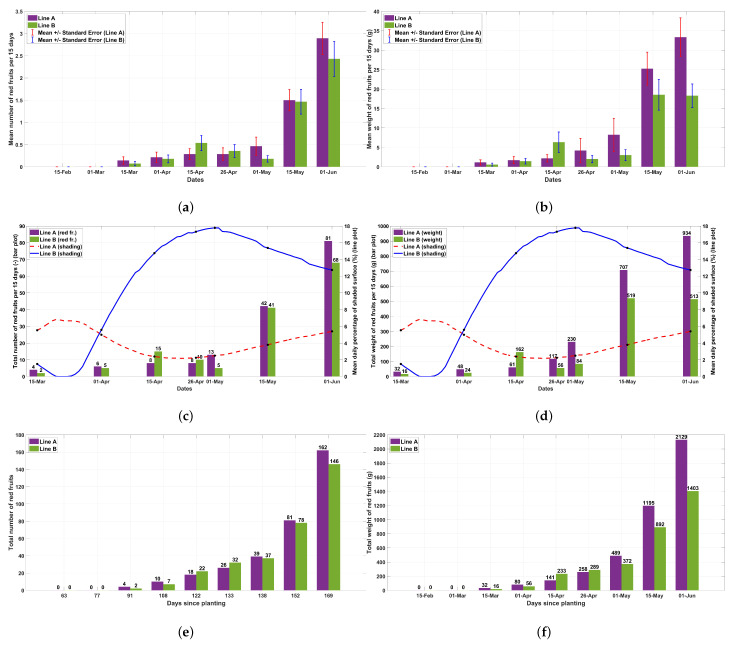
(**a**) Mean number of ripened (red) fruits during the experimental period (mean ± SE, n = 28). (**b**) Mean weight of ripened (red) fruits during the experimental period (mean ± SE, n = 28). (**c**) Total number of ripened fruits for each planting row combined with the mean daily percentage of the shaded surface. (**d**) Total weight of ripened fruits for each planting row combined with the mean daily percentage of the shaded surface. (**e**) Cumulative sum of the number of ripened fruits for each planting row. (**f**) Cumulative sum of the weight of ripened fruits for each planting row.

**Figure 7 plants-13-00768-f007:**
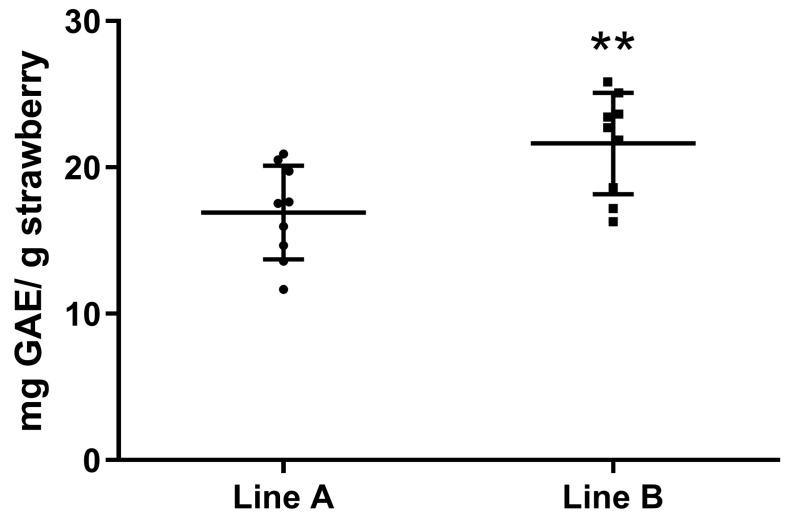
Results of total phenolic content in strawberries. The TPC in strawberries is expressed as mg of GAE per g of strawberries. Two asterisks (**) indicate statistically significant differences (*p* < 0.01) between the two strawberry planting lines.

**Figure 8 plants-13-00768-f008:**
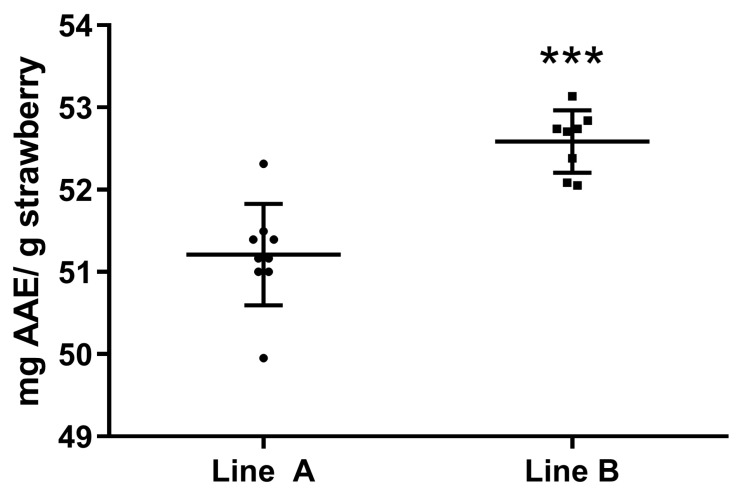
The results of the DPPH analysis in strawberries. The ability to scavenge DPPH in strawberries expressed as mg of AAE per g strawberry in different plot options. Three asterisks (***) indicate statistically significant differences (*p* < 0.0001) between the two strawberry planting lines.

**Table 1 plants-13-00768-t001:** Totals for the measured parameters and the percentage change between each parameter for lines A and B.

Parameter	Line A	Line B	Percentage Change (%)
Shading ($m^{2}$)	367.8	1099.6	+198.96
Number of Leaves	1579	1597	+1.14
Number of Flowers	181	169	−6.63
Number of Red Fruits	162	146	−9.88
Weight (g)	2123	1403	−33.91

## Data Availability

The data that support the findings of this study are available from the corresponding author upon reasonable request. The data are not publicly available due to confidentiality agreements with research participants.

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
