# Peer review of "Growth and Physiological Characteristics of Strawberry Plants Cultivated under Greenhouse-Integrated Semi-Transparent Photovoltaics"

_plants, 2024, doi:10.3390/plants13060768_

Round 1

Reviewer 1 Report

Comments and Suggestions for Authors

Congratulations to the authors because they have carried out a very interesting study on a very novel application in which I am very interested.

I would like to make some comments:

The scientific name of the plant used in this study should always be written in italics.

In order to improve the impact of the publication, it would be advisable to use keywords different from those that appear in the title.

Point 2.5 has a very generic title. It would probably be better to change to a more precise title, such as sample treatment or preparation, etc.

There is a parenthesis left on page 198.

The error bars of the variables represented are very wide and the number of samples is very high (n=28). It would be convenient to change SD to SE, dividing by the square root of the number of samples.

In table 1 the units of the Shading variable should be indicated.

Author Response

  • The scientific name of the plant used in this study should always be written in italics.

Reviewer's comment has been accepted and changes have been made in the title, and in lines 6 and 97. 

  • In order to improve the impact of the publication, it would be advisable to use keywords different from those that appear in the title.

The keywords have been changed.

  • Point 2.5 has a very generic title. It would probably be better to change to a more precise title, such as sample treatment or preparation, etc.

The reviewer's suggestion has been accepted and the title has been changed.

  • There is a parenthesis left on page 198.

Reviewer's comment has been accepted.

  • The error bars of the variables represented are very wide and the number of samples is very high (n=28). It would be convenient to change SD to SE, dividing by the square root of the number of samples.

We have replaced the standard deviation with the standard error, as suggested by the reviewer's comment. All corresponding figures have been updated accordingly.

  • In table 1 the units of the Shading variable should be indicated.

The units (m2) have been added next to the 'Shading' label in the first cell of Table 1.

"All of the changes have been highlighted in the manuscript"

Reviewer 2 Report

Comments and Suggestions for Authors

Dear Authors,

the overall quality of the manuscript is very high. I have appreciated your job but, there are some formal corrections to do.

About abstract, please remove the percentages of the results from the brackets and rephrases.

The cherachter size of the figures 4, 5 and 6 is to small and the graphs are not readable. Please, adjust it.

How many samples do you analyze for the regression on figure 7. Is very strange that R2 in perfectlly 1. Please, explicitate.

My general comments is that the manuscript can be accepted after minor revisions.

Author Response

  • About abstract, please remove the percentages of the results from the brackets and rephrases.

The requested changes have been made to the abstract. The percentages of the results have been removed from the brackets and rephrased accordingly.

  • The cherachter size of the figures 4, 5 and 6 is to small and the graphs are not readable. Please, adjust it.

The character size of Figures 4, 5, and 6 has been increased, as requested. All corresponding figures have been updated accordingly.

  • How many samples do you analyze for the regression on figure 7. Is very strange that R2 in perfectlly 1. Please, explicitate.

Its indeed a perfect linear regression, we analysed 3 different concentrations and the results can be provided upon reasonable request. 

"All of the changes have been highlighted in the manuscript"

Reviewer 3 Report

Comments and Suggestions for Authors

The manuscript is not novel because there are a lot of papers related to this topic.

The introduction is good, but the material and methods part have some information missing. How many replications of the greenhouses did the authors use for the study? The authors describe the radiation shading using an algorithm, but they didn’t measure it. The mean daily percentage of shaded surfaces shown in Figure 2 is very strange, and the authors didn’t explain these differences between consecutive months. Radiation measurements would be more useful to describe the real conditions than an algorithm. They didn't have a control greenhouse without photovoltaic system.

In line 174, the authors have written “Plants were covered with black plastic…” it is wrong, is the soil that was covered with plastic. What kind of drippers and how many were installed?

Results part: Figures axes labels have a small number size; it is difficult to read. Figures 4 a and b showed duplicate information. Figures 5 a and c, showed the same information, similarly in Figures 5 b and d. I will suggest diminishing the figures. It is not necessary to show the calibration curve in Figures 7 and 8.

In the discussion part, the authors explain the effect of temperature, but they showed only one temperature line, not two lines for the two scenarios. This section must be improved.

Author Response

  • The introduction is good, but the material and methods part have some information missing. How many replications of the greenhouses did the authors use for the study? The authors describe the radiation shading using an algorithm, but they didn’t measure it. The mean daily percentage of shaded surfaces shown in Figure 2 is very strange, and the authors didn’t explain these differences between consecutive months. Radiation measurements would be more useful to describe the real conditions than an algorithm. They didn't have a control greenhouse without photovoltaic system.

"Unfortunately, the University does not provide the possibility of two greenhouses. Therefore, a statement was added to the discussion section and lines 343 to 345, in order to clarify that the study focuses on the effect of different shading experienced by the plants of the two rows. The algorithm has been validated using radiation data collected from within the greenhouse (a phrase has been added in lines 145 and 146 ). The corresponding information has already been published, with the reference being on the line 146."

  • In line 174, the authors have written “Plants were covered with black plastic…” it is wrong, is the soil that was covered with plastic. What kind of drippers and how many were installed?

"We have corrected the sentence the reviewer indicated, while more information about the irrigation was added in lines 174 to 177."

  • Results part: Figures axes labels have a small number size; it is difficult to read. Figures 4 a and b showed duplicate information. Figures 5 a and c, showed the same information, similarly in Figures 5 b and d. I will suggest diminishing the figures. It is not necessary to show the calibration curve in Figures 7 and 8.

Figures axes labels have a small number size: "The character size of Figures 4, 5, and 6 has been increased to improve readability, as requested."

Figures 4 a and b showed duplicate information: "Figures 4a and b have been merged into one."

Figures 5 a and c, showed the same information, similarly in Figures 5 b and d. :  "Regarding Figure 5, we believe that the images give different information, since in Figures 5 a and b the authors refer to the mean value and standard error obtained from the plants of each line for each day when the measurements had been recorded (every 15 days), while in Figures 5 c and d the authors refer to the total number of leaves and flowers, respectively, for the corresponding days, together with the shading for each line. However, the y-axis labels of figures 5 & 6a, 5 & 6b, 5 & 6c, and 5 & 6d have been changed to make the information provided easier to understand."

It is not necessary to show the calibration curve in Figures 7 and 8. : "The calibration curves are a vital part of the chemical analyses performed and not including them would have been an omission on our part." 

  • In the discussion part, the authors explain the effect of temperature, but they showed only one temperature line, not two lines for the two scenarios. This section must be improved.

"The text of the first paragraph of the discussion (lines 305 to 310) has been revised in order to make clear our hypotheses about the effect of temperature"

"All of the changes have been highlighted in the manuscript"

Round 2

Reviewer 3 Report

Comments and Suggestions for Authors

After considering the explanations of the authors, I am disagreeing in accepting the manuscript. The algorithm used to calculate the shading surface was validated in another paper of the same group (Petrakis et al 2023). In that paper, they considered that the photovoltaic panels are opaque, but the pictures in Figure 1 showed that there is space between the cells. On the other hand, in the above-mentioned paper, the authors showed daily radiation evolution and we can observe that the heavy shadow is produced from 8:00 to 9:00 hours and 16:00 to 17:00 hours. Then, the percentage of shading in Figure 2 is not well calculated. For this reason, in my first revision, I asked for the radiation data measured during the essay.

With the radiation data information, the explanations of the results could be different. Perhaps the daily light integral in each crop row was different.

I suggest rewriting the results and discussion with this information.

Author Response

The semi-transparency of the photovoltaics installed in the greenhouse is a key aspect highlighted in our work. This characteristic is not only evident in the figures but is also reiterated throughout the manuscript, notably in the title. While it is acknowledged that in the previous study (Petrakis et al. (2023)), photovoltaics were considered opaque, our work addresses this disparity by incorporating adjustments to the calculated percentage of shaded area. Specifically, the results extracted from the algorithm have been refined using a scaling factor of 0.5108. This factor, derived from technical documentation accompanying the photovoltaic units, represents the semi-transparent nature of the modules (a phrase was added in lines 163-164). It quantifies the ratio of the active surface (comprising the solar cells causing shadow) to the total surface area of the photovoltaic module. This clarification is provided in lines 154 to 167 of the manuscript. Consequently, the shaded area percentage depicted in Figure 2 accurately reflects the semi-transparent nature of the photovoltaic panel. Lastly, it's important to highlight that the percentage of the shaded area calculated represents the daily mean value across the entire surface of each line. Consequently, when analyzing the results in the Results section and comparing them with the mean values across all plants within each line, we are confident in the validity of our findings.

The notable presence of shade between 8:00 to 9:00 and 16:00 to
17:00 hours, as observed in prior research, stems from the trajectory of shadow movement throughout the day intersecting with the pyranometers during these periods. This interaction is shown as reduced radiation levels observed by the pyranometers, which serve as a tangible indicator of shadow existence. Taking advantage of this phenomenon, we used the drop in radiation to evaluate our model's accuracy in forecasting shadow occurrences over these time frames. Through the proper setup of the pyranometer positions within the algorithm, we precisely identified instances when the shadow traversed these predetermined locations. As a result, when the shadow moved across the pyranometer, radiation levels decreased, validating the algorithm's ability to precisely identify shadow positions within the greenhouse. However, it is crucial to emphasize that, while the drop in radiation at these exact periods proves the presence of a shadow, it does not suggest that the strength of the shadow varies across different points of the surface during these hours. By developing this model, we can essentially detect changes in radiation caused by the effect of photovoltaics without having to measure radiation, which would be difficult for a relatively large surface area and two separate planting lines, such as those studied in this work.

Round 3

Reviewer 3 Report

Comments and Suggestions for Authors

After considering the explanations of the authors, I am disagreeing in accepting the manuscript. The algorithm used to calculate the shading surface was validated in another paper of the same group (Petrakis et al 2023). In that paper, they considered that the photovoltaic panels are opaque, but the pictures in Figure 1 showed that there is space between the cells. On the other hand, in the above-mentioned paper, the authors showed daily radiation evolution and we can observe that the heavy shadow is produced from 8:00 to 9:00 hours and 16:00 to 17:00 hours. Then, the percentage of shading in Figure 2 is not well calculated. For this reason, in my first revision, I asked for the radiation data measured during the essay.

With the radiation data information, the explanations of the results could be different. Perhaps the daily light integral in each crop row was different.

I suggest rewriting the results and discussion with this information.

Author Response

(The authors gave the same response as above.)
